# Expression, Regulation and Function of microRNA as Important Players in the Transition of MDS to Secondary AML and Their Cross Talk to RNA-Binding Proteins

**DOI:** 10.3390/ijms21197140

**Published:** 2020-09-27

**Authors:** Marcus Bauer, Christoforos Vaxevanis, Nadine Heimer, Haifa Kathrin Al-Ali, Nadja Jaekel, Michael Bachmann, Claudia Wickenhauser, Barbara Seliger

**Affiliations:** 1Institute of Pathology, Martin Luther University Halle-Wittenberg, 06112 Halle, Germany; marcus.bauer@uk-halle.de (M.B.); claudia.wickenhauser@uk-halle.de (C.W.); 2Institute of Medical Immunology, Martin Luther University Halle-Wittenberg, Halle 06112, Germany; christoforos.vaxevanis@uk-halle.de (C.V.); nadine.heimer@uk-halle.de (N.H.); 3Department of Hematology/Oncology, University Hospital Halle, 06112 Halle, Germany; haifa.al-ali@uk-halle.de (H.K.A.-A.); nadja.jaekel@uk-halle.de (N.J.); 4Helmholtz-Zentrum Dresden Rossendorf, Institute of Radiopharmaceutical Cancer Research, 01328 Dresden, Germany; m.bachmann@hzdr.de; 5Fraunhofer Institute for Cell Therapy and Immunology, 04103 Leipzig, Germany

**Keywords:** myelodysplastic syndrome, secondary acute myeloid leukemia, microRNA, RNA-binding protein, therapy

## Abstract

Myelodysplastic syndromes (MDS), heterogeneous diseases of hematopoietic stem cells, exhibit a significant risk of progression to secondary acute myeloid leukemia (sAML) that are typically accompanied by MDS-related changes and therefore significantly differ to de novo acute myeloid leukemia (AML). Within these disorders, the spectrum of cytogenetic alterations and oncogenic mutations, the extent of a predisposing defective osteohematopoietic niche, and the irregularity of the tumor microenvironment is highly diverse. However, the exact underlying pathophysiological mechanisms resulting in hematopoietic failure in patients with MDS and sAML remain elusive. There is recent evidence that the post-transcriptional control of gene expression mediated by microRNAs (miRNAs), long noncoding RNAs, and/or RNA-binding proteins (RBPs) are key components in the pathogenic events of both diseases. In addition, an interplay between RBPs and miRNAs has been postulated in MDS and sAML. Although a plethora of miRNAs is aberrantly expressed in MDS and sAML, their expression pattern significantly depends on the cell type and on the molecular make-up of the sample, including chromosomal alterations and single nucleotide polymorphisms, which also reflects their role in disease progression and prediction. Decreased expression levels of miRNAs or RBPs preventing the maturation or inhibiting translation of genes involved in pathogenesis of both diseases were found. Therefore, this review will summarize the current knowledge regarding the heterogeneity of expression, function, and clinical relevance of miRNAs, its link to molecular abnormalities in MDS and sAML with specific focus on the interplay with RBPs, and the current treatment options. This information might improve the use of miRNAs and/or RBPs as prognostic markers and therapeutic targets for both malignancies.

## 1. The Pathology of Myelodysplastic Syndromes

Myelodysplastic syndromes (MDS) are highly heterogeneous clonal hematologic disorders that are caused by unilineage or multilineage dysplasia accompanied by inefficient hematopoiesis resulting in peripheral cytopenia [1,2]. Multifactorial pathogenetic features are associated with diverse cytogenetic, molecular, and epigenetic alterations, which are accompanied by variable clinical presentations [3,4,5]. With an annual incidence of 3.5–10/100,000 the disease commonly occurs in elderly patients with a predominance in males [6]. While primary MDS is putatively linked to cigarette smoking, benzene exposure, and a positive family history, secondary MDS arise in a timely context of radiotherapy and chemotherapy [7]. MDS are characterized by peripheral blood cytopenia and a hypercellular bone marrow (BM) with a dysplastic, atypical hematopoiesis including an increased apoptosis rate [8]. Furthermore, MDS can transform into secondary acute myeloid leukemia (sAML). According to the revised World Health Organization (WHO) classification in 2017, de novo acute myeloid leukemia (AML) and sAML with MDS-related changes are distinct diseases, which significantly differ in terms of their biologic and clinical features [3,9]. In contrast, high risk MDS and sAML share phenotypic and genetic characteristics, including overlapping somatic mutations and chromosomal rearrangement alterations in a variety of genes involved in disease pathogenesis, such as RNA splicing, DNA methylation, ras signaling, and transcriptional regulation [10,11,12].

The typical genetic setting of MDS and sAML includes nonrandom cytogenetic and somatic mutations like balanced and unbalanced translocations, hypodiploidy, whole chromosome gains, deletions, and genomic alterations [13,14]. In addition, high throughput analyses, such as single nucleotide polymorphism (SNP) arrays, next generation sequencing (NGS), and RNA expression profiling identified a variety of molecular abnormalities, in particular in key genes and pathways, that are involved in the initiation and progression of MDS and lead to chromosomal and genetic instability, common transcriptional RNA splicing abnormalities and epigenetic changes [15]. Early paired analyses of MDS and sAML samples demonstrated the acquisition of novel mutations or an increased frequency of existing alterations implicating a role of these alterations in leukemic progression from MDS to sAML suggesting a multistep process of transformation from MDS to sAML [4]. However, the mechanisms contributing to progression into sAML have not yet been elucidated in detail [16].

To get insights into the biological properties of MDS, the Revised International Prognostic Scoring System (IPSS-R) is applied, which classifies patients into five groups based on the risk of mortality and development of sAML. This system comprises cytogenetics, bone marrow (BM) blast count, and the degree of peripheral cytopenia of respective lineages [17]. In this context the finding of karyotypic abnormalities, although not specifically named, justifies the diagnosis of MDS in those cases, where typical morphological stigmata are missing [2,3,18]. Furthermore, >40 different oncogenic events are related to MDS with more than 50% of patients harboring two or more somatic mutations [18]. Common driver mutations in MDS patients are described in genes affecting RNA splicing (splicing factor 3B subunit 1 (SF3B1)), serine and arginine rich splicing factor 2 (SRSF2), U2 small nuclear RNA auxiliary factor 1 (U2F1), zinc finger CCCH-type, RNA binding motif and serine/arginine rich 2 (ZRSR2), DNA methylation (Tet methylcytosine dioxygenase 2 (TET2), DNA (cytosin-5)-methyltransferase 3A (DNMT3A), isocitrate dehydrogenase (NADP(+)) 1/2 (IDH1/IDH2)), chromatin modification (additional sex combs-like 1 (ASXL1), enhancer of zeste 2 polycomb repressive complex 2 subunit (EZH2)), transcription regulation (RUNX family transcription factor 1 (RUNX1), (BCL6 corepressor (BCOR)) and DNA repair control (tumor protein 53 gene (TP53)) [5]. Therefore, not only cytogenetic findings, but also the number, specification and combination of mutations as well as the tumor mutational burden (TMB) influence the risk of leukemic progression and thus can be used as independent prognostic marker(s) and as candidate(s) for targeted treatment approaches [19,20,21,22,23].

Next to cytogenetic and molecular abnormalities, the composition and deregulation of the activity of immune cells in the microenvironment of hematopoietic stem cells (HSC) plays an important role in the biology and development of MDS and sAML. Indeed, evasion of adaptive immune surveillance is involved in the transformation of MDS to sAML. Major alterations of the tumor microenvironment (TME), especially concerning the connective tissue and the repertoire of immune cell subsets due to continuous inflammation and intrinsic tumor immunological response, were found in MDS [24,25]. In this context, a wide range of immune cell subtypes and their downstream signaling pathways participate in MDS pathogenesis and evolution. While immune activation is involved in the development of lower-risk MDS, escape from immune surveillance appears to contribute to high risk MDS. The latter mechanism includes a reduced frequency and functional exhaustion of CD4^+^ T cells [26]. an increased presence of T cell exhaustion markers and myeloid-derived suppressor cells (MDSC), and functional changes in NK cells [27]. Furthermore, programmed death ligand-1 (PD-L1), T cell immunoglobulin and ITIM domain (TIGIT), T cell immunoglobulin mucin-3 (TIM-3), and CD47 expression was frequently shown to be increased in MDS [26,28,29,30,31] whereas loss of heterozygosity (LOH) of HLA class I antigens only occurred in <10% of patients [27].

The mechanisms controlling disease progression and prognosis of MDS have not yet been identified in detail. However, during the last years, the composition of the BM microenvironment, in particular concerning the extracellular matrix and the repertoire of immune competent cells within the defective BM, its interaction with malignant HSC, and their monitoring via epigenetic and posttranscriptional control mechanisms including microRNAs (miRNAs) and long noncoding RNAs (lncRNAs) have been extensively studied [1,32,33]. Distinct miRNA expression profiles identified in sAML and MDS with either cytogenetic alterations or normal cytogenetics were associated with patients’ prognoses. Since miRNAs have been shown to be involved in immune escape [34], the evaluation of their involvement in MDS initiation and progression is urgently needed as their interference may have the power to influence treatment decisions and might serve as suitable therapeutic targets [25,35,36,37,38,39]. As a major point, RNA-binding proteins (RBPs) and miRNAs may compete for the same target sites or sequences, in which the binding of a RBP makes a miRNA recognition site more accessible to the RISC complex [40,41]. The interaction of miRNAs and RBPs is further underlined by the usage of similar enzymes for degradation of target transcripts. Therefore, this review will summarize (i) the strategies used to analyze miRNAs and/or RBPs in preleukemic disorders and sAML, (ii) the role of miRNAs in the context of immune escape and (iii) their regulation by RBPs, (iv) the association of miRNAs/RBPs profiles with cytogenetic aberrations in MDS/sAML and (v) their clinical significance as well as (vi) novel treatment options targeting miRNAs/RBPs in both diseases.

## 2. Characteristics of miRNAs

MiRNAs are small noncoding RNAs with a length of 18–25 nucleotides that regulate gene expression at the post-transcriptional level by destabilizing target transcripts or repressing the translation of their messenger RNA target [36]. The identification of the first miRNA dates back to 1993 with the discovery of lin-4, which has an antisense complementary sequence to several regions in the 3′ untranslated region (UTR) of the lin-14 gene [36,42] thereby regulating lin-14 expression [42]. In the last two decades, the biogenesis of miRNAs has been described and studies on miRNAs have substantially expanded [43]. MiRNA expression can be controlled by different mechanisms, such as regulation of transcription, epigenetics, alterations within the miRNA biogenesis machinery, and chromosomal abnormalities [44]. Changes in the miRNA profile have been linked to abnormalities in DICER, an RNAse III endonuclease required for miRNA biogenesis [45,46]. Furthermore, RNA editing involving specific enzymes of the adenosine/cytidine deaminase family, which trigger single nucleotide changes in primary miRNA and influence miRNA stability, maturation, and activity, deepen our knowledge about the molecular mechanisms underlying complex diseases [47].

MiRNAs play a key role in many physiologic and pathophysiologic processes and contribute to the control of e.g., cell proliferation, apoptosis, and differentiation. As major post-transcriptional regulators miRNAs are involved in a multitude of neoplastic and inflammatory diseases [24,48,49,50,51,52]. In this context, miRNAs can act as oncogenes or tumor suppressor genes, but can also have immune modulatory properties in different cancer types. The prominent role of miRNAs in neoplastic processes is strengthened by the fact that >50% of miRNA genes are located in cancer related regions. Thus, miRNAs might serve as valuable markers for diagnosis, prognosis, and as promising therapeutic targets for tumors [24,25,53].

Concerning miRNA expression in hematopoiesis, 77% of all miRNAs were identified in hematological malignancies, from which 18% are relevant in myeloid malignancies [54,55]. MiRNA expression is highly dependent on the cell type and developmental stage of myeloid cells [56]. Next to the deregulation of miRNAs in hematologic disorders, genetic variations in miRNAs, such as SNPs and single nucleotide variants (SNV) are able to disrupt the miRNA target interaction [57] or causing loss of miRNA expression [58]. These disrupting SNVs/SNPs might also play a role in MDS and/or sAML as shown for other cancers [59].

## 3. Features of RBPs

RBPs are involved in the post-translational control and play a key role in the regulation of protein expression, metabolism, transport and localization, stability, and degradation of various RNA molecules [60,61].

Changes in these processes affect the RNA life cycle, generate abnormal protein phenotypes and thus would result in the initiation and progression of tumors [62,63]. So far, more than 2000 RBPs have been identified [61]. Some of these RBPs have been shown to contribute to the development of hematological malignancies, which has been comprehensively summarized by various authors [64,65] as well as in this issue reviewed by Schuschel and co-authors [66]. There exists increasing evidence that cancer related alterations in RBPs are involved in miRNA biogenesis. MiRNA biogenesis, fate, and function [67] is modified and directed by dynamic interactions with RBPs [68] and miRNAs [35,36,68,69] Recently, a number of interactions of RBPs with miRNAs have been identified under various physiological and pathophysiological conditions. The significance of RBPs, in particular their roles in aberrant splicing and translation in hematologic malignancies, has been underlined by the comprehensive use of next-generation sequencing technologies [64,70,71,72].

## 4. Methodical Strategies and Specific Challenges Regarding miRNA and RBP Analyses in MDS and sAML

A number of different strategies were developed to identify differentially expressed miRNAs/RBPs in MDS and sAML and to evaluate their impact in disease initiation and progression. Regarding miRNAs next to computational approaches using different miRNA platforms, which allow the prediction of putative miRNAs binding to the 3′ UTR, 5′ UTR, and coding sequence (CDS) of target RNAs, additional technologies include high resolution microarrays, small RNA sequencing, and qPCR analyses of miRNA panels [73,74]. However, varying results were obtained employing the different microarray platforms (Agilent, Illumina, Affymetrix, Nanostring). One reasonable explanation for this finding might be that these platforms were developed to detect and quantify gene expression patterns, but might fail when dealing with miRNAs. These specific difficulties might be due to (i) the short nucleotide sequence of miRNAs thereby limiting the options for primer design, (ii) the need to differentiate between mature miRNAs from their precursors, as well as (iii) the multiplicity and dynamic range of miRNAs [75]. Recently, RNA sequencing as the most sensitive method has been successfully implemented for the identification of miRNAs. Comparison of qPCR and microarray technology with deep sequencing results using the Illumina platform demonstrated the best correlation between the Agilent qPCR platform and the deep sequencing [76,77]. A recent study of global miRNA sequencing in AML samples showed an advantage to microarray analysis, since it removes the bias due to its specificity regarding the 5p or 3p strand of the gene that might have functional relevance [78]. These technologies have been extended to single cell RNA sequencing. Furthermore, a combination of miRNA expression and bioinformatics prediction of mRNA target demonstrated a distinct miRNA profile of each step of hematopoiesis [79].

MiRNA analyses in cancer probes were successful in different settings including fresh frozen and formalin-fixed, paraffin-embedded tumor lesions, ascites, peripheral blood, urine, plasma, and isolated exosomes [80]. Microdissection might improve the purity, but will not completely avoid differences in the number of infiltrating cells. Therefore, single cell RNA sequencing is a challenging, but most promising approach for miRNA analyses. Furthermore, paired sequencing of miRNAs and mRNAs reveal a nongenetic cell-to-cell heterogeneity and will give insights into regulatory relationships of cellular pathways regulating miRNA expression [81]. In order to obtain candidate miRNAs relevant for defined activities, functional screenings using miRNA libraries can be also applied. In addition, a more targeted approach is the miRNA trapping by RNA in vitro affinity purification (miTRAP) analysis using the 3′ UTR/ 5′ UTR or CDS of the gene of interest followed by (small)RNA sequencing of the miTRAP eluate [82,83].

Determination of a specific miRNA expression profile including the functional role in the context of MDS and sAML is under debate as data are highly contradictory. This is partially due to the methodical differences and the source of material profiled for miRNA expression as studies included analysis on MDS blasts, bone marrow aspirates with and without stroma cells, peripheral blood specimen plasma, and extracellular microvesicles (MV). Recently, it has been shown that the RNA content of the EV cargo was more homogeneous when compared to total plasma which comprised a different set of deregulated miRNAs [84]. Even upon sorting of BM cells for the stem cell marker CD34, the CD34^+^ cell population is still heterogeneous, since immune cells committed in their early developmental phase to a particular lineage and endothelial cells also express CD34 [85]. Furthermore, blasts in MDS or AML represent a very heterogeneous population in terms of lineage commitment and some neoplastic blasts lack CD34 expression. A comparability of diverse studies is further aggravated by the fact that there exists a high variability in cut off values, normalization, and reference controls [86]. Despite these limitations, the different studies demonstrated significant differences in the miRNA profile between hematopoiesis from MDS patients and healthy controls.

Nevertheless, it should be noted that a proper investigation of the published data regarding (i) the miRNA source and (ii) the reference controls used is urgently needed to compare the different results. A compilation of the current data is the goal of this review.

Based on next generation sequencing technologies including single cell RNA-sequencing hundreds of RBPs and their associated RNAs have been identified, which allowed the prediction of RNA–protein interactions using deep learning approaches [87]. It highlights the importance of RBPs in splicing and translation [72]. Furthermore, other technologies were employed to identify RBPs. These include systematic analysis of ligands by exponential enrichment (SELEX), RNA immunoprecipitation (RIP)-based methods, cross-linking immunoprecipitation (CLIP), RNA pulldown assays, RNA affinity purification followed by mass spectrometry, and RNA screening within protein libraries [88]. In addition, a systematic screen using CRISPR/Cas targeting RNA-binding domains of a large series of RBPs can be used to identify interacting RBPs in malignancies as recently described for AML [89]. So far, these techniques were employed for the analysis of whole cell lysates. When the interaction with a specific subcellular compartmentalization was of interest, cell lysis of subcellular compartments was adapted [90]. However, as for miRNA identification, the cellular source should be considered next to the high dynamics of the protein–RNA complexes, which need to be investigated at a high temporal resolution [91].

## 5. miRNA Expression in MDS and sAML

Given their regulatory role, miRNAs have been evaluated in samples obtained from MDS patients, sAML patients, and healthy donors. Comparative studies revealed a differential miRNA expression pattern between these three groups [16,92] (Table 1). These studies were carried out to identify (i) diagnostic and prognostic markers for these diseases, (ii) biomarkers indicative of the progression from MDS to sAML, and (iii) novel therapeutic targets. Unfortunately, the studies do not completely result in a common consensus due to discrepancies concerning the analyzed samples, their cellular origin, and heterogeneity. This discrepancy might be attributed to the lack of the uniformity of the selected material, which ranges from miRNAs from BM mononuclear cells, surrounding mesenchymal cells, precursor cells, myeloid lineage cells to free and MV-bound circulating miRNAs (Figure 1A). Reviewing these studies, unique data pointing to a specific deregulation of miRNAs were not reported (Figure 1B). Noteworthy, some miRNAs were downregulated in MDS and sAML patients’ samples compared to healthy controls (Table 1A). One possible explanation was offered by Ozdogan and co-authors describing a downregulation of the ribonuclease III enzymes DICER and DROSHA in MDS specimen, which was even more pronounced in BM samples from sAML patients [46]. Indeed, a downregulation of these factors has a direct impact on the production and ultimate expression of mature miRNAs. However, these data are far away from their applicability with respect to the molecular profiling of miRNAs for their clinical application. Considering the limitations with regard to their comparability, special efforts were invested on liquid biopsies by analysis of free and MV-bound miRNAs from total blood, serum, and plasma. Using this approach, a higher miRNA content in MV of MDS patients compared to healthy controls was found [93] suggesting that the miRNA cargo of circulating MVs may contribute to the pathophysiology of this disease. Furthermore, the miRNA content differs between total plasma and EVs in MDS versus healthy controls [84].

Differentially expressed miRNAs in MDS compared to healthy controls, the cellular source, clinical relevance, chromosomal location, cytogenetic aberrations, frequency of aberrations (in %), are summarized according to Haase 2008 [101]. In addition, RBPs targeting the miRNAs investigated as well as other known targets were given by cross referencing with the RBPDB database (http://rbpdb.ccbr.utoronto.ca/).

Gaining increased insights in miRNAs recorded in MDS and/or sAML, it became obvious that they function either as oncogenes or tumor suppressors thereby affecting a broad range of disease-associated processes including cell proliferation, survival, differentiation, epigenetic regulation, self-renewal, disease progression, and (chemo)therapy resistance [53]. Based on the published literature and own results, miRNAs can be grouped into four main categories: (i) MDS-related miRNAs, (ii) AML-related miRNAs and (iii) miRNAs that are either dysregulated in both diseases, or (iv) miRNAs that are implicated in the progression from MDS to AML. By evaluation of the published data, we found that independent miRNAs accumulated with close to 75% in total (sAML-related 33%, MDS-related 56%), and a high percentage of these miRNAs overlapped in both diseases (Figure 2). Interestingly, bioinformatics analyses to identify miRNA targets demonstrated that a broad number of candidate targets were members of oncogenic signaling pathways as well as molecules involved in apoptosis, differentiation, and DNA repair [46]. However, by reducing the number of the main candidates, the results from the different studies remained contradictory. Regarding miR-181a and miR-181b, their high expression was associated with the inability of AML blast differentiation and highly increased probability of MDS progression to sAML in one study [102], while collective miRNA signature data, others describe a better prognosis of MDS patients when these miRNAs were upregulated [103]. In contrast, concordant data are presented for miR-15/-16. In detail a double knockout of the two miR-15/-16 loci in mice caused the development of AML [104]. These results could be directly translated into humans, since BM probes from patients with MDS transforming into sAML or from patients with sAML expressed lower levels of both miRNAs when compared to corresponding samples from MDS patients [84,105].

A special issue concerning deregulated miRNAs in MDS and sAML are immune modulatory miRNAs (im-miRNAs). The duality of the immune system’s role as a regulator of immunity and hematopoiesis highlight the significance of these miRNAs involved in the initiation of immune responses as well as of immune cell proliferation. In this context, inflammation related miRNAs (e.g., miR-146a, miR-155, miR-23b, miR-9, miR-223) and miRNAs involved in alterations of the immune cell composition (miR-150, miR-181a, miR-181b, miR-223, miR-17-92 family) as well as miRNAs controlling the expression of immune modulatory molecules, such as miR-155 and miR-330, have been shown to be deregulated in one of the three categories described above [106]. Based on the literature research, the miRNA expression levels, the cellular source together with their chromosomal location and clinical relevance in MDS and sAML are summarized in Table 1.

In MDS, mutations of genes involved in the pre-mRNA splicing (SF3B1, SRSF2, U2AF1, and ZRSR2) are frequently detected and occur in more than 50% of all MDS patients [70,71,107] suggesting that the spliceosomal dysfunction plays a major role in disease pathogenesis. Consecutively, aberrantly spliced target mRNAs cause a deregulation of intracellular pathways influencing the initiation and the course of the disease [108,109]. These regulatory circuits are complex layers controlling cancer initiation and progression [110] and their role in MDS and sAML will be discussed.

## 6. Functional Relevance of Differentially Expressed miRNAs in MDS and AML

Due to the fact that a single miRNA can target several hundred mRNAs, alterations in the miRNA expression can affect a variety of cellular processes involved in disease initiation and progression [111]. Mechanisms, by which miRNAs promote MDS/sAML are mediated by different signaling pathways including inflammation [112,113,114,115]. Therefore, targets of miRNAs were classified concerning their function and participation in signaling cascades using bioinformatics annotation tools. Taken together, around 73% of the miRNAs described target genes were involved in transcriptional and translation regulation, control of immune responses, tumor immune surveillance, and tumorigenicity as shown in Figure 3. Approximately one third of these miRNAs play a key role not only in MDS/sAML, but also in the progression of solid tumors like lung and breast cancer [116,117]. In total, 25% of the selected miRNAs are associated with sAML by targeting the PI3K/AKT/mTOR, MAPK and/or JAK-STAT signaling pathways. Experimental downregulation of miRNAs involved in PI3K/AKT/mTOR cascades resulted in an increased proliferation, reduced apoptosis, and an upregulation of immune checkpoint molecules, like HLA-G, accompanied by an immune evasion of malignant cells [118]. The expression pattern of some miRNA candidates showed a correlation with other signal transduction pathways including TGF-ß, Wnt, and p53, while other deregulated miRNAs target general transcription factors, such as NF-κB, which disrupts normal NF-κB signaling resulting in neoplastic transformation [119]. Several reports underline the importance of miRNA regulation of the NF-κB pathway for progression of AML in general, but also of MDS and the promotion of leukemic phenotypes [120,121,122]. Furthermore, the toll-like receptor signaling, which is well documented in MDS, leads to constitutive activation of NF-KB, which could be fine-tuned by miR-125a [115].

## 7. Clinical Relevance of miRNAs in MDS and AML

Deregulated miRNA expression profiles in samples with clinical follow up could be correlated with disease progression and was used for patients’ risk assessment and prediction of therapy response in hematologic malignancies including MDS (Table 1B) [35]. Using different high throughput technologies, the miRNA expression profiles were assigned to the IPSS-R and characterized [38,39,123,124].

Again, these studies significantly differed with regard to the cellular source investigated. While some studies used unsorted BM cells [100,123] others examined mononuclear cells [38,39] [94,96,124,125] or CD34^+^ BM cells [126], respectively. It is noteworthy that each investigative approach has its own advantages and disadvantages. For instance, although the CD34^+^ BM cell population is still heterogeneous, it captures the MDS-specific blast population closer than other studies demonstrating intrinsic changes in the miRNA expression in HSC of MDS patients. On the other hand, the analyses of unsorted BM cells can deliver more information of the microenvironment and the enhanced erythroid differentiation. Independent of the sample source, significant differences in the miRNA expression patterns between healthy controls, MDS, and sAML patients were identified (Table 1). Furthermore, functional miRNA expression clusters/profiles varied in the distinct MDS subgroups and therefore might be of prognostic relevance, whereas individual miRNAs can have a comparable expression in all MDS subtypes and sAML (Table 2) [39]. Since some of the differentially expressed miRNAs appear to be of prognostic value upon categorization patients into high and low risk groups of MDS and sAML patients’ progression from MDS to sAML [100,127] miRNAs play a context-dependent role in MDS/AML. For example, downregulation of miR-155 has a prognostic value for MDS, since its upregulation is associated with a bad prognosis, whereas this miRNA has no phenotypic effect in MLL-rearranged de novo AML, but in FTL3-ITD-driven pathogenesis [128,129].

The same miRNAs appear to contribute to the clinical outcome of MDS patients along with other miRNAs that do not necessarily have a direct connection to immune processes. This observation underlines changes in malignant cells and in the cellular components of the TME demonstrating the complexity of these deregulations during disease progression.

## 8. Predictive Value of miRNA Expression from Patient’s Plasma

Next to miRNA analyses in cells, the use of plasma and serum as liquid biopsies offers new roads of diagnostic value [130]. Significant differences of single miRNAs overexpressed in plasma of MDS versus healthy donors (HD) (miR-144, miR-96-5p, miR-493) or vice versa (miR-206, miR-34b, miR-651) were found. Interestingly, none of them were identical to those detected in BM tissues from MDS and HD (Table 1A). Analysis of the miRNA cluster (miR-let7a, -16, -25, -144, -451, -651, -655) allowed the categorization of patients into different prognostic groups. Let-7a, miR-16, miR-191, and miR-199a were identified as markers linked to a shorter progression-free survival (PFS) and overall survival (OS) [131,132]. Furthermore, miRNAs could be detected in MV [133], in particular in exosomes released from tumor cells with a content distinct from the miRNA population of the host cell [134,135]. It is suggested that these MVs mediate cell-to-cell communication through miRNA transfer thereby modulating the functional properties of hematopoietic cells [136]. Indeed, the miRNA cargo of MDS-derived MVs including miR-10a and miR-15a are involved in the increased viability of CD34^+^ cells, but also in their clonogenicity suggesting an important role in the maintenance of clonal hematopoiesis [136]. In addition, MV tracking might have the potential for early detection of disease recurrence [137].

## 9. Correlation of microRNAs with Cytogenetic Features in MDS and SAML

Genomic abnormalities, such as deletions, polyploidy, translocations, SNPs, and complex chromosomal aberrations frequently occurring in hematologic disorders might be linked to an altered miRNA expression pattern [138]. This suggests also a diagnostic and prognostic potential of these molecules as biomarkers. Data from a study on 2072 MDS patients [101] classifying the most common single chromosomal abnormalities demonstrated that many deregulated miRNAs were localized in these regions (Table 1). However, some deregulations of miRNA cannot be explained by chromosomal abnormalities, even in patients exhibiting phenotypes with several chromosomal deletions/additions. Consequently, the differential expression patterns of the surrounding cells as well as epigenetic modifications have to be considered to contribute to the miRNA deregulation.

A variety of nonrandom cytogenetic aberrations with prognostic value have been reported. Trisomy 8 and deletions of a part or the total chromosomes 5, 7, and 20 are common events in MDS [139]. Some abnormalities were linked to the diagnosis of MDS without evidence of dysplasia [2]. Many patients with del(5q) show consistent clinical features.

The karyotype changes affect the expression of genes as well as miRNAs. The two miRNAs miR-145 and miR-146a have been implicated in the pathogenesis of the 5q- syndrome [140,141] while other cytogenetic aberrations e.g., del(7q) and del(20q), affect miRNA expression levels, which could be used for risk stratification of the patients’ outcome [142]. MiR-145 and miR-146 expression are reduced in MDS samples leading to an upregulation of the toll-interleukin-1 receptor domain containing adaptor protein (TIRAP), which is associated with an impaired innate immune signaling. Furthermore, both miRNAs contribute to the pathogenesis of del5q MDS by inappropriate activation of innate immune signals. Since the constellation of cytogenetic aberrations in combination with changes of miRNA expression levels play a critical role, karyotype-dependent miRNA studies are urgently required to get better insights into miRNAs relevant for the pathogenesis of MDS. Table 1 summarizes the most common cytogenetic aberrations and MDS/sAML-associated miRNAs [139,143].

## 10. Crosstalk of miRNAs and RBPs in MDS and SAML

The interplay between miRNAs and RBPs at the 3′UTR has been shown to modulate target gene expression. Furthermore, binding of RBPs near miRNA target sites could regulate miRNA function by either affecting miRNA binding or by switching the secondary RNA structure. Recently, protein interactors of more than 70 miRNA precursors have been identified in human tumor cell lines of distinct origin including cell lines from hematopoietic disorders [67].

The targets of the investigated miRNAs described in this review were validated using the database for experimentally validated miRNA–target interactions miRTarBase (http://mirtarbase.cuhk.edu.cn). As representatively shown in Table 1B/C, the miRNA-RBP interplay might be linked to the pathogenesis of MDS and sAML [67,144,145,146]. In order to increase the validity of these findings, only targets identified and validated by luciferase reporter assays, Western blot, and/or qPCR were taken into account. Furthermore, the resulting targets were cross referenced with the RBPDB database (http://rbpdb.ccbr.utoronto.ca/) to identify all known RBP targets for the miRNA candidates (Table 1).

Mutations in spliceosome components occur in more than 50% of MDS samples with SFB1, SRSF2, U2AF1, and ZRSR2 as the most frequent mutated splicing factor genes [70]. A consequence of these mutations is the aberrant splicing, which had a significant effect on miRNA expression: covering all the intentional hotspots within SF3B1, SRSF2, and U2AF1 to a downregulation of canonical miRNAs compared to wild type was found. The most downregulated miRNAs were the tumor suppressive miRNAs of the let-7 family, miR-423 and miR-103a [147] (Table 3).

RNA modification in both coding and noncoding RNAs by methylation of N6-methyladenosine (m6A) is catalyzed by the RNA methyltransferase-like (METTL) METTL3, METTL14, and METTL16 that can be removed by the two demethylases, the fat mass- and obesity-associated gene (FTO), and the Alk B homolog 5 (ALKBH5), which interact with m6A-binding proteins, such as the YTH domain-containing family protein 1 (YTHDF1) and the insulin growth factor 2 binding protein 1(IGF2BP1) [154,155,156,157,158] affecting protein expression in physiological and pathological conditions.

## 11. Treatment Options of mRNA Binding Modulators in MDS and SAML

Manipulation of the expression levels of mRNAs targeting intracellular pathways involved in physiologic and/or pathologic processes might represent novel powerful therapeutic treatment options [159,160]. A snapshot of the current developmental status of targets of miRNA- and RBP-based modulators and connected therapeutics and their targets in MDS and sAML is summarized in Table 4.

## 12. Therapeutic Possibilities of miRNAs in MDS and SAML

Different strategies using miRNAs as therapeutic agents have been reported. One option is in case of miRNA overexpression the miRNA inhibition using antisense oligonucleotides, so called antagomiRs, containing complementary sequences of endogenous miRNAs. Potential targets of this technology are miR-29, miR-34, and miR126-5p [171,172,173]. In the case of oncogenic miRNA downregulation, re-expression of these miRNAs using expression vectors or transfection of double-stranded miRNAs can be performed [174,175]. Possible candidates for this therapeutic approach in MDS and sAML are miR-22, miR-33, miR-134, and miR-181 [176,177,178,179] as well as miRNAs targeting TIM-3 and CD47, known to be frequently overexpressed in MDS and/or sAML. These include e.g., miR-34, miR-15a/-16, miR-138a-5p, miR-149-3p, miR-455-5p miR-498, and miR-330 (Table 5) [180,181]. Some of these miRNAs play an important role in hematologic malignancies and/or to negatively regulate the expression of TIM-3 and/or CD47 in hematopoietic cells [180,181,182,183,184]. Interestingly, miR-155, miR-34, and miR-133a target both TIM-3 and CD47. Other options targeting miRNAs are the implementation of enzyme inhibitors involved in pathways affecting miRNA expression levels [185].

Currently, no miRNA-based therapeutics for MDS and sAML are available. However, it is encouraging that the first small-interfering RNA (siRNA) drug Onpattro ^®^ (patisiran), a therapy for the rare hereditary disease transthyretin-mediated amyloidosis in adult patients, was approved by the Food and Drug Administration (FDA) and the European Medicines Agency (EMA) [196,197]. Furthermore, a number of miRNA vaccine therapies are currently in the clinical development or in phase I/II trials, while others were halted due to severe side effects [159]. The possibilities in the implementation of miRNAs as treatment option are rapidly progressing in a wide range of diseases alone or in combinations with other therapeutic agents affecting genetic and epigenetic targets. It is noteworthy that miRNA drugs can be applied in a comparatively simple manner in hematologic malignancies reaching the BM via blood circulation. In this context, it is suggested to develop the implementation of miRNAs targeting e.g., CD47 or TIM-3 for the treatment of MDS and/or sAML diseases.

## 13. Treatment Approaches Targeting the Spliceosome in MDS and SAML

There exists recent evidence that common heterozygous hotspot mutations in splicing factors can be targeted by spliceosome inhibitors and splicing modulators and have a therapeutic value in the treatment of splicing factor mutant myeloid malignancies [198,199]. Since let-7 family members, miR-423 and miR-103a are known to bind to the 3´UTRs of SF3B1, SRSF2, and U2AF1, respectively, leading to a downregulation of their expression, overexpression of these miRNAs might represent a promising approach [147]. Other possibilities are the orally available splicing modulator H3B-8800 targeting SF3B1 that induced cell death in spliceosome-mutant cancer cells [166]. Treatment of SRSF2 mutant mice with the spliceosome inhibitor E7107 resulted in a substantial reduction of leukemic burden [167]. Currently, first clinical trials using H3B-8800 (NCT02841540) and E7107 (NCT00499499) are performed. Another feasible candidate is the inhibitory molecule E7820 targeting the auxiliary RNA splicing factor RBM39 [146].

## 14. Targeting RNA Methylation in MDS and SAML Treatment

Since METTL3 has been shown to be an essential gene for the growth of normal hematopoietic cells as well as AML blasts, this enzyme serves as therapeutic target [200]. Indeed, several possible small-molecule inhibitors of METTL3 have been developed and promising substances have been identified [154]. Another feasible target is the major m6A demethylase FO known to cause a dramatic suppression of proliferation in glioma and to promote the differentiation and apoptosis of AML blasts [156]. Different FTO inhibitors like meclofenamic acid, rhein, and FB23-2 have been tested, but so far, no clinically employed drugs are currently available [168,169,170].

## 15. Treatment Options of mRNA Binding Modulators in MDS and SAML

As in many diseases, MDS and MDS-related AML patients show clear differences in the miRNA expression pattern compared to healthy individuals, which might have prognostic and predictive value (Table 1). It seems obvious that the manipulation of the expression levels targeting intracellular pathways involved in (patho)physiologic processes might provide potentially powerful treatment options [159,160]. In experimental AML models, the delivery of miRNA-29b decreased the leukemic growth and improved the expression of target genes due to miRNA-mediated downregulation [201]. Since the miRNA regulation in MDS patients can be impaired due to cytogenetic aberrations with subsequent loss of miRNAs, the loss of miR-146a and miR-145 in patients with a 5q-syndrome causes a better response to Lenalidomide therapy [140]. MiRNAs have been also employed in MDS/AML as predictors to hypomethylating agents (HMA) in order to distinguish between HMA resistance and sensitivity [202]. Indeed, the oncogenic miR-21 and miR-181 expression correlated with the response to HMA in MDS patients [173,203]. while oncogenic expression of miR-331 and miR-181 were associated with a rather diverse response to HMA. In addition, lower expression levels of miRNA-126 and the miR-29 family, both targeting the DNA methyltransferases (DNMT) 1 and 3, predicted reduced response rates and poor outcome of patients [204,205]. These data suggest that monitoring of miRNA expression before and during treatment might help to predict primary or secondary resistance to HMA [206]. However, even if the methodology may be successfully applied, there are numbers of hurdles that need to be overcome, whereby in the context of hematologic neoplasia uniform miRNA profiling might be the most critical issue. Here, as a general rule, the respective miRNAs have to target the pathogen causal hematopoietic precursor cell population to obtain valid results.

## 16. Conclusions

MiRNAs simultaneously targeting intracellular pathways are involved in a broad range of (patho)physiologic processes. They are critical regulators of hematopoiesis and their aberrant expression has been associated with various types of leukemia [35,36,39] Their functional role in the context of MDS and sAML is under debate due to methodical differences and the source/material profiled for miRNA expression in both diseases [38,39,94,96,100,123,124,125,126]. Sorted CD34^+^ BM cells still comprise a heterogeneous cell population and not all neoplastic blasts in MDS and sAML express CD34 [85]. However, this cellular source is nonetheless the best characterized compartment for miRNA studies. In this context, it seems obvious that the monitoring of the miRNA expression profiles may provide diagnostic, prognostic, and predictive markers for therapy response. Furthermore, miRNAs could serve as potential drug targets for novel miRNA-based therapies. The manipulation of the expression of specific miRNAs or their targets could be used as a potential powerful therapeutic option.

However, there are numbers of hurdles that need to be overcome. Beyond all barriers and limitations, a proper monitoring of the impact of aberrant miRNA expression in MDS and sAML might lead to novel targeted therapeutic strategies. These therapeutic strategies have to target the pathogenic causal hematopoietic precursor cell source. It is important to emphasize that even after identification of promising miRNAs further challenges, such as RNA instability, optimal dose for patients’ injection, and drug delivery have to be solved [207]. In addition, pharmacologic trials concerning pharmacokinetics and pharmacodynamics are challenging prerequisites that have to be performed before clinical trials can be carried out. Taken together, it has to be assumed that even though miRNA drugs in hematological malignancies can be applied in a comparatively simple manner reaching the BM via blood circulation, the therapeutic implementation of miRNA will still be a long-term goal.

## Figures and Tables

**Figure 1 ijms-21-07140-f001:**
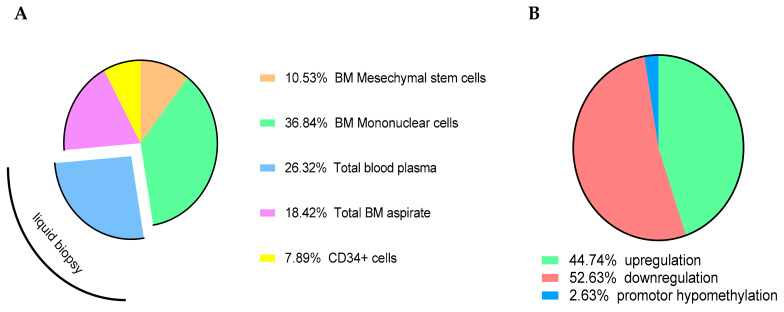
Cellular source and miRNA dysregulation. (**A**) MiRNA expression data based on various different materials/sources used including total BM or blood aspirates, BM mononuclear cells, or isolated CD34^+^ cells. (**B**) Reviewing these studies, a specific deregulation of miRNAs could not be identified.

**Figure 2 ijms-21-07140-f002:**
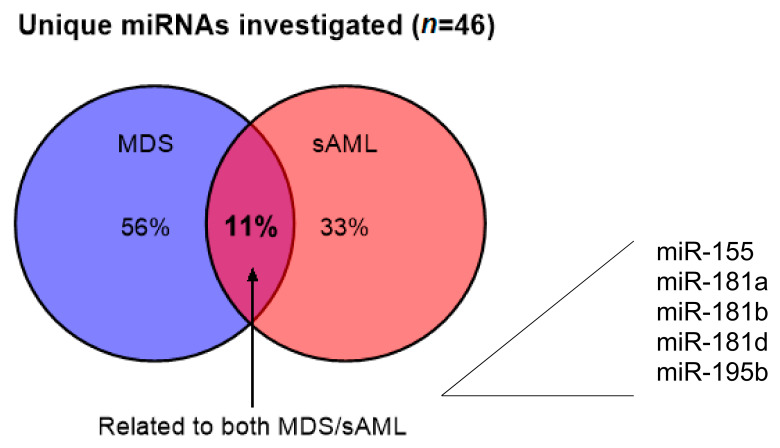
Frequency of dysregulated miRNAs in MDS and sAML. Close to 89% of the investigated miRNAs (*n* = 46) showed a deregulation of miRNA in MDS or sAML when compared to healthy controls, in which 33% are sAML-related and 56% are MDS-related. Approximately 11% of deregulated miRNAs overlapped in both diseases (11%).

**Figure 3 ijms-21-07140-f003:**
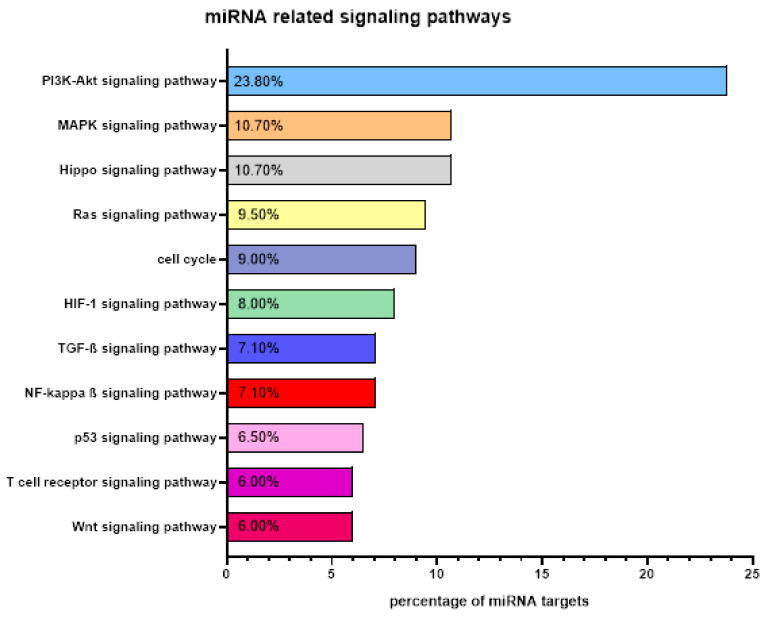
Targets of miRNAs in MDS and AML. MiRNAs can target various intracellular pathways that are necessary for a broad range of biological and pathophysiological processes including transcriptional and translational regulation, control of immune responses, and tumor immune surveillance. The major miRNA-related signaling pathways in MDS and sAML are presented.

**Table 1 ijms-21-07140-t001:** Differentially Expressed miRNAs, their Regulation, their Targeted RNA-Binding Proteins (RBPs) and Clinical Parameters in MDS and sAML.

A: miRNAs with Diagnostic Role									
miRNA	Disease	Cellular Source	Regulation *^1^	Chromosomal Location	Cytogenetic Aberration	% (*n* = 2072)	Target RBPs	Other Targets	Ref.
let-7a	MDS	BM mononuclear cells	↓	9q22.32	−9q	1.10	EWSR1	HMGA2, KRAS, HRAS, HMGA1, RRM2	[94]
miR-16-5p	MDS	Total blood plasma	↓	3q25.33, 13q14.2	inv/t(3q), −13/−13q	2.0/1.9	n.a.	BCL2, VEGFA, CCNE1, FGF2, CCND1	[95]
miR-27a-3p	MDS	Total blood plasma	↑	19p13.12	No	-	n.a.	FOXO1, SP1, ZBTB10, IFNG HOXA10	[95]
miR-30a-3p	MDS	BM mesenchymal stem cells	↓	6q13	No	-	n.a.	CDK6, SLC7A6, TMEM2, THBS1, CYR61	[46]
miR-30d-5p	MDS	BM mesenchymal stem cells	↓	8q24.22	8	8.40	n.a.	TP53, SNAI1, EZH2, BCL9, NOTCH1	[46]
miR-124a	MDS	BM mononuclear cells	↓	8p23.1	8	8.40	PTBP1, HNRNPA2B1	CDK6, STAT3, SLC16A1, EZH2, AMOTL1	[94]
miR-150-5p	MDS	Total blood plasma	↑	19q13.33	No	-	n.a.	MZB, EGR2, P2RX7, ZEB1, MUC4	[95]
miR-155	MDS	BM mononuclear cells	↓	21q21.3	+21, −21	2.2/1.6	n.a.	CEBPB, TO53INP1, SOCS1, BACH1, INPP5D	[94]
miR-182	MDS	BM mononuclear cells	↓	7q32.2	−7/−7q, t(7q)	11.1, 1.1	n.a.	RECK, MITF, FOXO1, BCL2, PDCD4	[94]
miR-195b-5p	MDS/sAML	BM mononuclear cells	↑	17p13.1	−17/−17p	2.00	n.a.	CCND1, WEE1, BCL2, E2F3, CDK6	[96]
miR-199a-5p	MDS	Total blood plasma	↑	1q24.3, 19p13.2	+1/ +1q	1.80	n.a.	HIF1A, SIRT1, MAP3K11, ERBB2, CAV1	[95]
miR-200c	MDS	BM mononuclear cells	↓	12p13.31	−12, −12p	1.3, 1.2	n.a.	ZEB1, ZEB2, TUBB3, BMI1, KRAS	[94]
miR-205-5p	MDS	Total BM aspirate	↑	1q32.2	+1/+1q	1.80	n.a.	ERBB3, VEGFA, ZEB1, ZEB2, E2F1	[97]
miR-222-3p	MDS	BM mesenchymal stem cells	↓	Xp11.3	No	-	n.a.	CDKN1B, KIT, PTEN, TIMP3, ETS1	[46]
miR-342-5p	MDS	BM mononuclear cells	↓	14q32.2	No	-	n.a.	NAA10	[94]
miR-451a	MDS	Total blood plasma	↓	17q11.2	−17/−17p	2.00	n.a.		[95]
miR-595	MDS	CD34^+^ cells	↓	7q36.3	−7/−7q, t(7q)	11.1, 1.1	n.a.		[98]
miR-4462	MDS	BM mesenchymal stem cells	↑	6p21.2	No	-	n.a.		[46]
**B: miRNAs with Prognostic Role**									
**miRNA**	**Disease**	**Cellular source**	**Prognostic marker *^2^**	**Chromosomal Location**	**Cytogenetic Aberration**	**% (*n* = 2072)**	**Target RBPs**	**Other targets**	**Ref.**
let-7a-3	MDS	BM mononuclear cells	Adverse *^3^	12q13.31	−12, −12p	1.3,1.2	EWSR1	HMGA2, KRAS. HRAS, HMGA1, RRM2	[98]
miR-021	MDS	BM mononuclear cells	Adverse	17q23.1	−17/−17p	2.00	n.a.	PTEN, RECK, PDCD4, BCL2, TPM1	[94]
miR-27a-3p	MDS	Total blood plasma	Good	19p13.12	No	-	n.a.	FOXO1, SP1, ZBTB10, IFNG HOXA10	[95]
miR-124a	MDS	BM mononuclear cells	Adverse	8p23.1	8	8.40	PTBP1, HNRNPA2B1	CDK6, STAT3, SLC16A1, EZH2, AMOTL1	[94]
miR-126	MDS	BM mononuclear cells	Adverse	9q34.3	−9q	1.10	n.a.	VEGFA, CRK, IRS1, PIK3R2, SPRED1	[94]
miR-146b-5p	MDS	BM mononuclear cells	Adverse	10q24.32	No	-	n.a.	MMP16, TRAF6, IRAK1, IL6, PDGFRA	[94]
miR-150-5p	MDS	Total blood plasma	Good	19q13.33	No	-	n.a.	MZB, EGR2, P2RX7, ZEB1, MUC4	[95]
miR-155	MDS	BM mononuclear cells	Adverse	21q21.3	+21, −21	2.2/1.6	n.a.	CEBPB, SOCS1, TO53INP1, BACH1, INPP5D	[94]
miR-195b-5p	MDS/sAML	BM mononuclear cells	Adverse	17p13.1	−17/−17p	2.00	n.a.	CCND1, WEE1, BCL2, E2F3, CDK6	[96]
miR-199a-5p	MDS	Total blood plasma	Good	1q24.3, 19p13.2	+1/+1q	1.80	n.a.	HIF1A, SIRT1, MAP3K11, ERBB2, CAV1	[95]
miR-223-3p	MDS	Total blood plasma	Good	Xq12	No	-	n.a.	LMO2, IGF1R, NFIA, RHOB, FBXW7	[95]
miR-451a	MDS	Total blood plasma	Good	17q11.2	−17/−17p	2.00	n.a.	n.a.	[95]
**C: miRNAs Indicative of Progression from MDS to sAML**									
**miRNA**	**Disease**	**Cellular Source**	**Regulation *^4^**	**Chromosomal Location**	**Cyto-genetic Aberration**	**% (*n* = 2072)**	**Target RBPs**	**Other Targets**	**Ref.**
miR-145	MDS	CD34^+^ cells	↓	5q33.3	−5q, −5, t(5q)	15.1, 3.3, 1.2	n.a.	IRS1, POU5F1, FSCN1, KLF4, IGF1R	[99]
miR-146a	MDS	CD34^+^ cells	↓	5q33.3	−5q, −5, t(5q)	15.1, 3.3, 1.2	ELAVL1	IRAK1, TRAF6, EGFR, NFKB1, STAT1	[99]
miR-181a-5p	MDS	Total BM aspirate	↑	1q32.1, 9q33.3	+1/ + 1q, −9q	1.8, 1.1	n.a.	BCL2, ATM, DDX3X, PRKCD, CDKN1B	[100]
miR-181b-5p	MDS	Total BM aspirate	↑	1q32.1	+1/ + 1q	1.80	n.a.	TIMP3, BCL2, CYLD, TCL1A, CREB1	[100]
miR-181d-5p	MDS	Total BM aspirate	↑	19p13.12	No	-	n.a.	BCL2, HRAS, MGMT, RAP1B, MEG3	[100]
miR-195b-5p	MDS/sAML	BM mononuclear cells	↑	17p13.1	−17/−17p	2.00	n.a.	CCND1, WEE1, BCL2, E2F3, CDK6	[96]
miR-199b-5p	MDS	Total BM aspirate	↑	9q34.11	−9q	1.10	n.a.	JAG1, HES1, PODXL, ERBB2, SET	[100]
miR-451a-5p	MDS	Total BM aspirate	↓	17q11.2	−17/−17p	2.00	n.a.	n.a.	[100]
miR-486-5p	MDS	Total BM aspirate	↓	8p11.21	8	8.40	n.a.	OLFM4, IGF1R, DOCK, CIT, ARHGAP5	[100]

*^1^ ↓: downregulation in MDS compared to HC, ↑: upregulation in MDS compared to HC; *^2^ adverse: higher expression levels related to worse prognosis groups, good: higher expression levels related to better prognosis groups; *^3^ indirect effect due to promotor hypomethylation; *^4^ ↓: downregulation in MDS patients progressing to sAML, ↑: upregulation in MDS patients progressing to sAML.

**Table 2 ijms-21-07140-t002:** Expression Profile of Selected miRNAs in MDS Subtypes and SAML.

miRNA	HC	MDS with Del (5q)	MDS-RS	MDS-MLD	MDS-EB1	MDS-EB-2	AML-MLD
	(*n* = 9)	(*n* = 6)	(*n* = 5)	(*n* = 6)	(*n* = 7)	(*n* = 10)	(*n* = 9)
miR-10a	↑	↑	↑	↓	↓	↓	↓
miR34a	↑	↑	↑	↑	↑	↓	↑
miR322-3p	↑	↑	↑	↑	↑	↑	↑
miR-378	↓	↓	↓	↑	↓	↓	↓

Comparison of miRNAs expression in CD34^+^ blasts derived from different MDS subgroups: MDS with isolated del(5q) (MDS with del(5q)), MDS with ringed sideroblasts (MDS-RS), MDS with multilineage dysplasia (MDS-MLD), MDS with excess of blasts 1 and 2 (MDS-EB-1 and -2), as well as AML with multilineage dysplasia (AML-MLD). Adapted from Merkerova et al. 2011 [39].

**Table 3 ijms-21-07140-t003:** miRNAs Targeting Genes Frequently Mutated in MDS and/or sAML.

Mutated Genes	miRNAs Identified	Cells/Tissues Analyzed	References
TET 2	miR-22 let-7 family	MDS Cytopenia Macrophages	[148,149,150] [148] [149]
ASXL1/IDH1	miR-21 miR-125a miR-143/-145 miR-146a	Various cell types	[151]
SF3B1/SRSF2/42AF1	let-7 family miR-103a miR-423	MDS	[147]
RUNX1	miR-9-5p	AML	[152]
TP53	miR-661	MDS	[153]

List of miRNAs targeting genes commonly mutated in MDS, along with the tissues analyzed.

**Table 4 ijms-21-07140-t004:** A Snapshot of the Current Developmental Status of miRNA-Based and Connected Therapeutics in Diseases.

Context	Target	Importance in MDS Pathogenesis	Disease for Which the Drug Is Being Developed	Drug Name and Company	Mechanism of the Drug	Clinical Relevance	Clinical Trial	Phase of Study	Ref.
**miRNA**	miR-124a	Downregulation	Ulcerative colitis	Abivax, ABX464	Enhanced splicing of miR-124	Yes	NCT03093259	Phase 2	[161]
			Ulcerative colitis	Abivax, ABX464	Enhanced splicing of miR-124	Yes	NCT03760003	Phase 2	
			Crohn’s disease	Abivax, ABX464	Enhanced splicing of miR-124	Yes	NCT03905109	Phase 2	
	miR-155	Upregulation	Cutaneous T-cell lymphoma (CTCL), mycosis fungoides (MF), CLL, DLBCL-ACB type, adult T-cell leukemia/lymphoma (ATLL)	Cobomarsen, MRG-106	Inhibitor of miR-155	Yes	NCT02580552	Phase 1	[162]
			Mycosis fungoides	Cobomarsen, MRG-106	Inhibitor of miR-155	Yes	NCT03837457	Phase 1	
	miR-200c	Downregulation	Osteogenesis	pSil-miR200c plasmids	Locally delivering plasmid DNAs encoding miR-200c NAs	Yes	NCT02579187	Withdrawn	[163]
	miR-21	Downregulation	Alport syndrome	Regulus, RG-102	Inhibitor of miR-21	No	NCT03373786	Phase 1	[164]
	miR-16-5p	Upregulation	Malignant pleural mesothelioma, NSCLC *	MesomiR	mirR-16 based mimic	No	NCT02369198	Phase 1	[165]
**Spliceosome**	SF3B1 *	Frequently mutated in MDS (>30%)	MDS, CMML *, AML	H3B-8800	Inhibitor	Yes	NCT02841540	Phase 1	[166]
	SRSF2 *	Frequently mutated in MDS (>15%)	Cancer	E7107	Inhibitor	Yes	NCT00499499	Phase 1	[167]
	RBM39 *	Auxiliary RNA splicing factor	MDS, AML, CLL *	E7820	Inhibitor	Yes	-	Phase 0	[81]
**mRNA methylation**	METTL3 *	RNA methylase	Hematologic neoplasm	Not yet identified	Inhibitor	Yes	-	Phase 0	[154]
	FTO *	RNA demethylase	Brain tumors	Meclofenamic acid	Inhibitor	Yes	-	Phase 0	[168]
			-	Rhein	Inhibitor	Yes	-	Phase 0	[169]
			AML	FB23-2	Inhibitor	Yes	-	Phase 0	[170]

Summary of clinical trials with potential relevance in MDS and sAML employing miRNA-based drugs targeting miRNAs, published in the context of inflammatory, solid, and lymphatic neoplasms. * NSCLC: non small cell lung cancer, SF3B1: Splicing Factor 3B Subunit 1, CMML: Chronic Myelomonocytic Leukemia, SRSF2: Serine/Arginine-rich Splicing Factor 2, RBM39: RNA-Binding Protein 39, CLL: Chronic Lymphocytic Leukemia, METTL3: N6-adenosine-methyltransferase 70kDa Subunit, FTO: Fat mass and obesity-associated protein.

**Table 5 ijms-21-07140-t005:** miRNAs Identified to Target the Checkpoint Molecules TIM-3 and CD47.

Immune Checkpoint	miRNA	Reference
TIM-3	miR-15a/-16	[186]
	miR-34a *	[183]
	miR-133a-5p *	[182]
	miR-149-3p	[187]
	miR-155	[188,189]
	miR-330-5p *	[180,190]
	miR-455-5p *	[184]
	miR-498 *	[181]
CD47	miR-34	[191]
	miR-133a *	[192]
	miR-155	[193]
	miR-200a	[194]
	miR-326	[191]
	miR-708	[195]

* miRNAs identified and/or functionally analyzed in cell lines or samples from hematopoietic malignancies. MiRNAs identified to target the MDS-relevant, immune checkpoints TIM3 and CD47. Regulating miRNAs found in cell lines or samples of hematopoietic malignancies are marked with an asterisk (*).

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
