# Peer review of "Expression, Regulation and Function of microRNA as Important Players in the Transition of MDS to Secondary AML and Their Cross Talk to RNA-Binding Proteins"

_ijms, 2020, doi:10.3390/ijms21197140_

Round 1

Reviewer 1 Report

This is an accurate and comprehensive account of the role of miRNA and RNA-binding proteins in MDS.

  1. Please include a discussion on how miRNAs regulate expression of immune checkpoints such as TIM3 and CD47, both of which are implicated in MDS. Anti-TIM3 and anti-CD47 have shown significant activities in combination with hypomethylating agents.
  2. For table 2 on page 12 and in the entire manuscript, please use the 2016 WHO classification scheme.
  3. Please include a discussion and table correlating miRNA expression with various genetic aberrations in MDS, such as those involving SF3B1, TET2, DNMT3A, IDH1/2, TP53, RUNX1 and ASXL1. These aberrations are prognostically important.
  4. There is no mention of chronic myelomonocytic leukemia. This should be included and compared with other MDS in terms of miRNA expression.

Author Response

Changes regarding the comments and suggestions of rev. no. 1:

  1. As suggested, we included a discussion (Lines 428-447) of microRNAs directed against TIM-3 and CD47 and added a new Table (Table 5). The microRNAs were listed and information regarding the expression/detection in hematopoietic malignancies/cell lines is marked.
  2. As suggested, the 2016 WHO classification scheme was used during the whole manuscript including table 2.
  3. We included a novel Table 1 (extended the information) discussing the genes with hotspot mutations in MDS/sAML in view of microRNA deregulation. This table includes the major microRNAs binding to the respective targets exhibiting a high frequency of genetic variations. This issue was also further discussed. Furthermore, we added a new table (Table 3) containing miRNAs targeting genes frequently mutated in MDS and/or sAML.

Concerning comment no. 4 we did not include and mention data of chronic myelo-monocytic leukemia. This is based on the fact that this disease has different clinical properties than MDS, despite it has similar genetic alterations compared to unique MDS. Otherwise, we should report different MDS and similar diseases, such as CMML.

Reviewer 2 Report

The authors present an update on miRNA role in MDS and AML. They have summarized a complex literature in a well-presented review. There are few concerns that the authors should address.

1.there are some grammatical errors

2.MDS abbreviation stands for both the singular and the plural form

3.All gene names should be written out in full

4.Figure 2. Overlapping miRNAs could not be related to disease progression.

5.The legend of table 1 is not appropriate.  The table represents more than “miRNA regulation in MDS and AML”

6.Table 1. The miRNAs should be listed in ascending order, so the reader can easily find what he is looking for. Moreover, could be useful to report the validated target of the listed miRNAs.
